# Transport regimes of a split gate superconducting quantum point contact in the two-dimensional LaAlO$_3$/SrTiO$_3$ superfluid

Holger Thierschmann [1], Emre Mulazimoglu [1], Nicola Manca [1], Srijit Goswami[1,2], Teun M. Klapwijk[1,3] & Andrea D. Caviglia[1]

One of the hallmark experiments of quantum transport is the observation of the quantized resistance in a point contact in GaAs/AlGaAs heterostructures. Being formed with split gate technology, these structures represent in an ideal manner equilibrium reservoirs which are connected only through a few electron mode channel. It has been a long standing goal to achieve similar experimental conditions also in superconductors. Here we demonstrate the formation of a superconducting quantum point contact (SQPC) with split gate technology in a two-dimensional superconductor, utilizing the unique gate tunability of the superfluid at the LaAlO$_3$/SrTiO$_3$ interface. When the constriction is tuned through the action of metallic split gates we identify three regimes of transport: First, SQPC for which the supercurrent is carried only by a few quantum transport channels. Second, superconducting island strongly coupled to the equilibrium reservoirs. Third, charge island with a discrete spectrum weakly coupled to the reservoirs.

[1] Kavli Institute of Nanoscience, Faculty of Applied Sciences, Delft University of Technology, Lorentzweg 1, 2628 CJ Delft, The Netherlands. [2] QuTech, Delft University of Technology, Lorentzweg 1, 2628 CJ Delft, The Netherlands. [3] Physics Department, Moscow State University of Education, Moscow 119991, Russia. These authors contributed equally: Holger Thierschmann, Emre Mulazimoglu. Correspondence and requests for materials should be addressed to H.T. (email: h.r.thierschmann@tudelft.nl) or to A.D.C. (email: a.caviglia@tudelft.nl)

Ever since the seminal experiments by van Wees et al.[1,2] on a quantum point contact formed with split gates in a semiconductor heterostructure, it has been a great experimental challenge to achieve similar experimental conditions also in superconductors[3]. This is generally desirable because the split gates enable quantum transport experiments within one and the same electronic system, without the need to combine different material systems. Such structures ideally represent equilibrium reservoirs which are connected only through a quantum constriction with a set of conducting channels, each of which has a certain transmission probability, as envisioned in the Landauer-Büttiker picture of quantum transport. With superconductors such conditions were accomplished only in atomic scale mechanically tunable break junctions of conventional superconducting metals, but here the Fermi wavelength is so short that it leads to a mixing of quantum transport with atomic orbital physics[4].

Split gates further allow for convenient in situ control of sample properties such as Fermi wavelength and the shape of the confinement potentials for charge carriers. Various attempts have therefore been made to combine the desired gate-tunability of the low electron density semiconductor with the use of conventional superconductors. However, these hybrid devices have introduced, compared to the GaAs/AlGaAs normal quantum transport case, the very important and yet very difficult to control influence of the interface between the two dissimilar materials[5]. This makes the results dependent on the complexities of the proximity effect and thus complicates their interpretation.

In principle, a new path has become available when it was discovered that in the two dimensional electronic system (2DES) at the LaAlO$_3$–SrTiO$_3$ (LAO–STO) interface superconductivity becomes suppressed when the electron density $n$ is reduced below a critical value $n_c$, for example, by means of a gate voltage[6,7]. The Fermi-wavelength $\lambda_F$ in this system can be as large as 30–50 nm[8] and ballistic transport in the normal state has been demonstrated[8,9]. The superconducting coherence length is about $\xi = 100$ nm[10,11]. This corresponds to spatial dimensions which are commonly achieved with present day lithography techniques. The creation of a superconducting quantum point contact (SQPC) with split gates in LAO/STO should therefore be within reach [cf. Fig. 1a]. In contrast to previous results on hybrids and mechanical break junctions, which used conventional bulk superconductors, it is to be expected that the two-dimensionality of the LAO/STO superfluid will play a significant role for the outcome of such an experiment. This approach can also offer insight into the nature of superconducting pairing at oxide interfaces. Unconventional pairing was recently suggested[12,13] in light of a number of experimental observations, including strong spin orbit coupling[14], co-existence of ferromagnetism and superconductivity[15–17], indications for electron pairing without macroscopic phase coherence[8,18–20] and a nontrivial relation between the critical temperature $T_c$ and charge carrier density $n$[6,20].

Here we present experiments that demonstrate the formation of a SQPC with split gates in the LAO/STO superfluid. We find that the quantum constriction undergoes different regimes of transport when tuned through the action of the split gates. While for more open configurations the supercurrent is carried by a single transport mode, a charge island is formed when the constriction becomes pinched off. The island energy spectrum is dominated by Coulomb repulsion and it exhibits a superconducting ground state when it is coupled more strongly to the reservoirs while for weaker coupling discrete electronic energy states appear.

## Results

**Split gate control of a quantum constriction.** Our samples are fabricated following the procedure described by Goswami et al.[7] (see Methods section and Supplementary Table 1). The measurements discussed in the main text were obtained from a single device, further measurements from a second sample are provided in the Supplementary Note 7. Figure 1b presents a false color atomic force microscope image of the device layout. The metallic split gates (yellow) L and R cover the full width of the 5 μm wide 2DES (blue), except for a 150 nm region at its center. Transport experiments are performed in a current bias configuration (unless stated otherwise) at temperature $T_{base} < 40$ mK. The resistively measured transition to the superconducting state is observed at $T_c \approx 100$ mK (see Methods section and Supplementary Fig. 2). Because of gate history effects, we carry out the experiment by putting electrode L on a fixed gate voltage ($V_L = -1$ V) to ensure depletion and tune the constriction by only varying the voltage $V_R$ applied to gate R (see Supplementary Note 6).

We expect the following scenario: When $V_R$ is changed toward negative values the charge carrier density $n$ gets reduced locally underneath the gate and gets closer to the critical density $n_c$ at which superconductivity becomes suppressed. At a certain gate voltage $V_R = V_c$ the condition $n = n_c$ is reached and a supercurrent can flow only through the constriction between the tips of the gates, thus forming a weak link between the

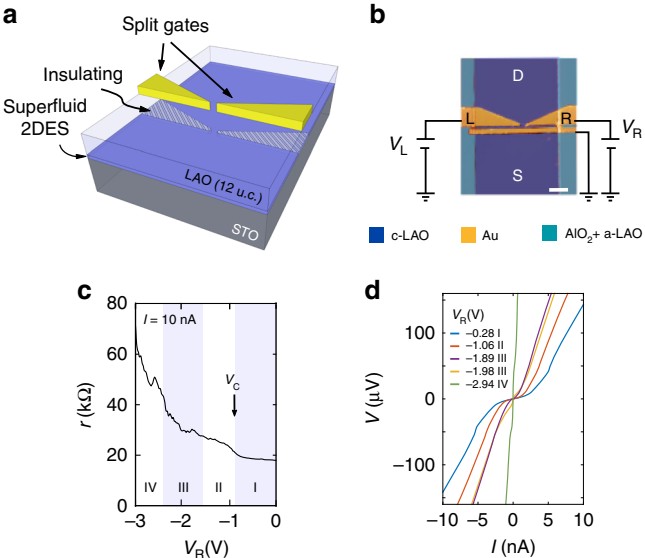

Fig. 1 Two-dimensional superconductor with split gates. **a** Generalized sketch of the device. At the interface between the STO substrate and the 12 unit cell (u.c.) layer of crystalline LAO (c-LAO) the superconducting 2DES (blue) is formed, which can be tuned insulating (shaded blue) locally under the split gates (yellow), thus forming a superconducting constriction. **b** False color atomic force microscope image showing the device layout. The potential of the split gates (yellow) L and R is controlled with the voltages $V_L$ and $V_R$, respectively. $V_L$ is kept at $-1$ V. The conductive 2DES is formed in regions with c-LAO (blue). In areas which are protected with an AlO$_2$ hard mask LAO growth is amorphous (a-LAO, turquoise). The thin gate spanning the channel is not used in the experiments. It is therefore kept at ground potential. Scale bar = 1 μm. **c** High bias ($I = 10$ nA) differential resistance $r$ as a function of $V_R$. I, II, III, and IV indicate the different regimes of transport (see text). $V_c$ denotes the formation of the constriction. **d** I–V curves for different gate voltages $V_R$ with $V_L = -1$ V. I–IV refer to the different regimes of transport indicated in Fig. 2. For regime III two curves are shown with a high and a low zero bias resistance, respectively

superconducting reservoirs. Outside this weak link, under the gates, the system acts as an insulator[6]. The number of transport modes available in the weak link is determined by its effective width. The constriction width is reduced when $V_R$ is further decreased and therefore the number of transmission channels decreases which is expected to lead to a step-wise reduction of the critical current $I_c$[3]. For $V_R \ll V_c$ transport will be dominated by a low transmissivity and the current is pinched off.

In order to study this scenario, we record a series of V–I curves and vary $V_R$ from 0 to −3 V. Panoramic overviews of the results are given in Fig. 2a, b in color plots. Figure 2a presents the differential resistance $r = dV/dI$ and Fig. 2b shows the differential conductance $g = dI/dV$ with the current $I$ and voltage drop $V$ on the vertical axis, respectively. It can be seen that the constriction undergoes four different regimes of transport (labelled I–IV in the figures) as $V_R$ is varied from 0 to −3 V. Representative I–V curves from each regime are shown in Fig. 1d. Regime I (ranging from $V_R = 0$ to −0.9 V) corresponds to the open current path configuration with $V_R > V_c$. A sharp peak in $r$ is visible at $I = \pm 5$ nA, labelled $I_P$ in Fig. 2a, which is reminiscent of a critical current $I_c$. Correspondingly, a dip occurs in $g$ at $V_P = \pm 44$ µV (Fig. 2b). At $V_R = V_c \approx -0.9$ V the critical density $n_c$ is reached. Here $I_P$ drops significantly because the current path becomes confined. At high currents in Fig. 2a, as shown for $I = 10$ nA in Fig. 1c, this point of confinement is apparent in a step increase in $r$, similar to the well-known behavior in semiconductor heterostructures[21]. It marks the transition to regime II ($V_R = -0.9$ to −1.6 V). In this regime $I_P$ decreases when $V_R$ is reduced indicating the gate tunable weak link. In regime III ($V_R = -1.6$ to −2.4 V) regions of high resistance at zero bias appear and disappear periodically. As we will show

below, this can be attributed to the emergence of a conductive island which dominates transport through the constriction. In regime IV ($V_R = -2.4$ to −3 V) the device always exhibits a high resistance at zero bias. Figure 2b reveals that this regime is controlled by conductance diamonds (CDs) (indicated with dashed lines).

**Superconducting quantum point contact**. Let us start with the weak link regime II. Here we observe a rounded supercurrent and an excess current $I_{exc} \approx 1$ nA [Fig. 3a]. $I_P$, reminiscent of the critical current $I_c$, changes from 3.7 to 3.0 nA [Fig. 3b] when $V_R$ is varied. The voltage $V_P = 44$ µV [cf. Fig. 3c] can be related to the superconducting gap $V_P/2 \approx \Delta \approx 22$ µeV, which is compatible with the value inferred from the resistively measured $T_c$, $\Delta_{Tc} = 1.76 k_B T_c = 15$ µeV. The high bias conductance $g_n$ [Fig. 3d] is of the order of half the quantum of conductance, changing with $V_R$ from 0.6 to 0.47 ($2e^2h^{-1}$) [20–28 kΩ]. As shown by Monteiro et al.[22] the phase correlation length in our 2DES is about 170 nm, whereas the lithographically determined channel-width is about 150 nm. It is therefore reasonable to interpret the data from a quantum transport perspective. For low carrier densities the Fermi wavelength $\lambda_F$ is several tens of nanometers[8]. This and the relatively low value of $g_n$ suggest that we have only a few modes with a finite transmissivity in the channel. With increasing $V_R$ we do not observe the expected quantum transport step-like features in $g_n$, although the trace in Fig. 1c is obviously not monotonous. This is not surprising because for currents larger than $I_c$ we approach the high bias regime where the conductance steps are known to quickly disappear[9,23]. In order to extract the transmissivity of the weak link we calculate from $I_{exc}$ and $g_n$ the barrier strength $Z$ as a function of $V_R$ using the BTK-formalism for an S–S interface[24]. $Z$ is related to the normal state transmission probability $\tau$ by $\tau = (1 + Z^2)^{-1}$. In this manner we obtain $Z \approx 0.8$ and, correspondingly, $\tau \approx 0.6$ [Fig. 3e]. Comparison with the measured $g_n$ thus suggests a total mode conductance of $2e^2h^{-1}$, such that $g(\tau = 0.6) = 0.6 \times 2e^2h^{-1}$, close to the measured values. If we follow recent experiments by Gallagher et al.[9] who observed $e^2h^{-1}$ modes in a normal state QPC, we could also consider only one mode with a higher transmissivity. However, this would require a re-analysis of the excess current based on an unconventional order parameter.

If we continue the discussion in the conventional picture, for a SQPC with perfect transmission ($\tau = 1$) Beenakker and van Houten[3] found that the critical current is given by $I_c = Ne\Delta(\hbar)^{-1}$, where $N$ was chosen to represent the number of spin degenerate modes (which contribute each $2e^2h^{-1}$ to the normal conductance). Using this relation and including the obtained $\tau$ as a prefactor, we can calculate the maximum supercurrent expected for our device, which yields $I_c \approx 3$ nA. This is in good agreement with the measured $I_P$, as can be seen in the bottom panel in Fig. 3e. For comparison we also plot the expected $I_c$ for a diffusive junction[25], which clearly gives much smaller values. The critical current $I_c \approx 3$ nA implies a Josephson coupling energy $E_J = 6.2$ µeV. This is comparable to the bath temperature, $k_B T_{base} = 3.4$ µeV. Therefore, as for the few-mode atomic scale point contacts[26,27], the supercurrent is rounded.

**Zero-dimensional charge island with discrete states**. Let us now turn to the regime of CDs, regime IV. Figure 4a presents a detailed measurement of $g$ in this region. Note that this measurement was carried out in a voltage bias configuration. We observe a series of CDs whose size $E$ on the (vertical) voltage axis is of the order of 80–150 µV. In gating experiments with non-superconducting materials, for instance in narrow semiconductor channels or graphene nano ribbons, CDs are known to occur in the low density limit, at the metal–insulator transition, because of puddles of charge carriers which form due to small inhomogeneities in the

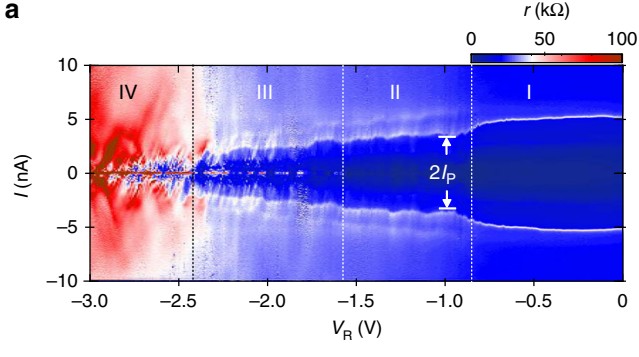

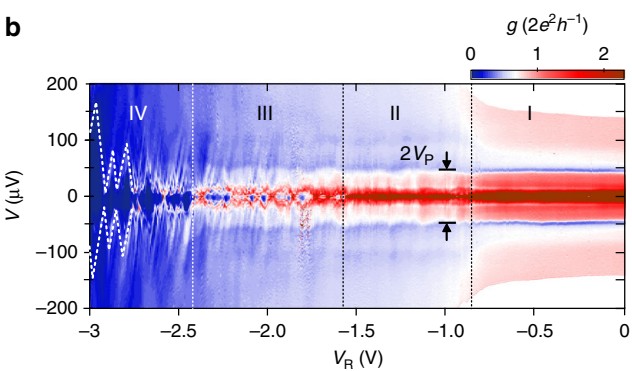

Fig. 2 Transport regimes in the constriction. **a** Differential resistance $r$ versus current $I$ for gate voltage $V_R = 0$ to −3 V. Gate L is kept at $V_L = -1$ V. The four regimes of transport I to IV are indicated. $I_P$, reminiscent of the critical current $I_c$ of the weak link, is indicated. **b** Same data as in **a** but with voltage drop $V$ on the vertical axis and differential conductance $g$ represented by the color scale. $V_P$ denotes the voltage drop at $I_P$. Dotted lines in regime IV indicate conductance diamonds

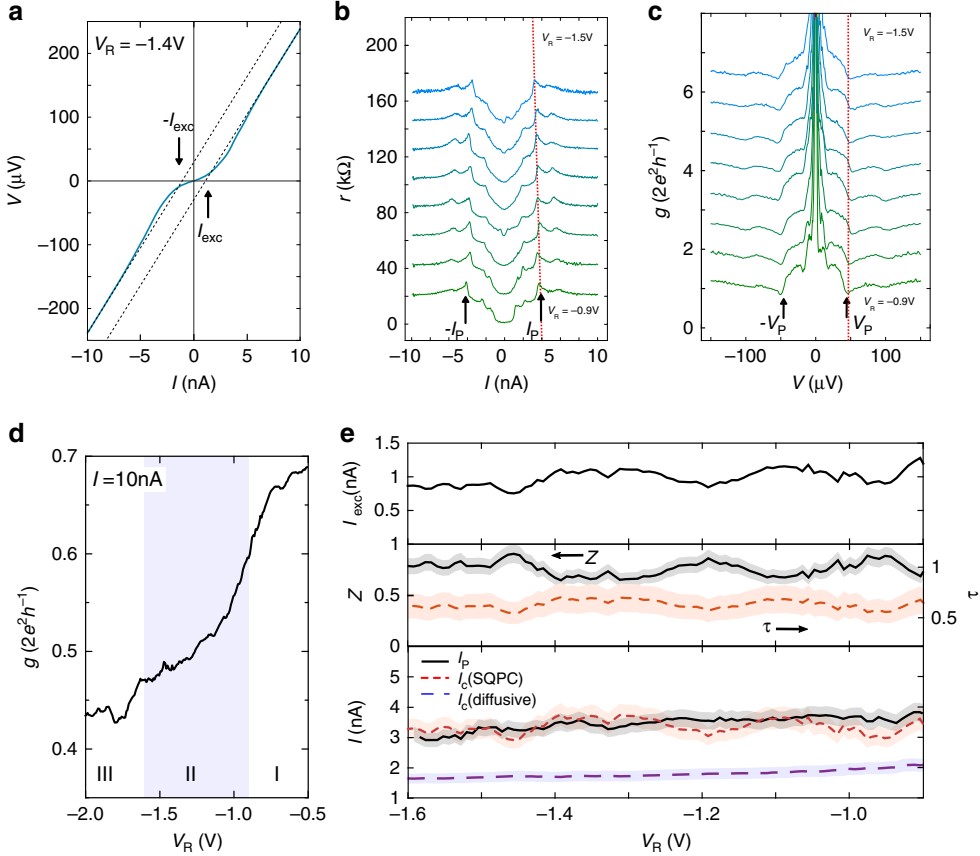

Fig. 3 Superconducting quantum point contact in regime II. **a** Representative $V$–$I$ curve in regime II ($V_R = -1.4$ V). The rounded supercurrent is clearly visible. The excess current is denoted $I_{exc}$. **b** $r$ versus $I$ for different $V_R$. $I_P$, inferred from the sharp peak in $r$, is indicated. The change of $I_P$ with $V_R$ is highlighted by a dashed line as guide-to-the-eye. The curves are offset by 10 kΩ. **c** $g$ versus $V$ for different $V_R$. $V_P \approx 44$ μV is indicated with arrows and a dashed line as guide-to-the-eye. The curves are offset by 0.2 ($2e^2h^{-1}$). **d** $g$ for $I = 10$ nA. **e** top panel: $I_{exc}$ as a function of $V_R$. Middle panel: Barrier strength $Z$ (solid, black line) and normal state transmission $\tau$ (dashed, red line) as obtained from BTK. Bottom panel: $I_P$ (solid, black line) and critical current $I_c$ as expected for a SQPC (short, red dashes). For comparison, $I_c$ expected for a diffusive junction with conductance $g$ is shown (blue dashes). Shaded regions indicate the standard deviation

potential landscape, thus leading to quantum dot-like transport behavior[28–33]. A similar behaviour is also conjectured to occur in 2D superconductors around the transition from the superconducting to the insulating state[34,35]. From this analogy we infer that in regime IV the superfluid inside the constriction is at the transition to full depletion. Similar observations have been reported recently also by Prawiroatmodjo et al.[36]. The size of the CDs directly reflects the addition energy $E$ that has to be paid in order to change the island occupation number and thus, to enable transport. $E$ is composed of various contributions of which the most dominant ones typically are the Coulomb charging energy $U = Ne^2(2C_\Sigma)^{-1}$ [with $C_\Sigma$ being the total capacitance of the island and $N$ the number of charges to be added or removed] and the energy level quantization due to quantum confinement $\delta\varepsilon$. For quantum dots in LAO/STO[18,19,37], Coulomb contributions are small because the STO substrate exhibits an extremely large dielectric constant $\epsilon_r = 25,000$ at low $T$ and for small electric fields[38] which suppresses Coulomb repulsion. For our device, however, the fields originating from the split gates can not be neglected[22]. We have performed simulations of the dielectric environment in the region surrounding the constriction using finite element analysis (see Supplementary Note 2). Our results indicate that the geometry of the gates leads to a strong field focusing effect which reduces $\epsilon_r$ in the constriction such that Coulomb repulsion becomes relevant. The numerical simulations yield charging energies of $U \approx 100$ μeV for an island with ~50 nm radius, compatible with our experiment. The data in Fig. 4a, further

shows signatures of transport through excited states originating from quantum confinement, as can be seen from the fine structure of conductance lines parallel to the diamond edges between two adjacent diamonds [green arrows in Fig. 4a][31,39]. This allows us to estimate $\delta\varepsilon \approx 10$–20 μeV, which would lead to an island size of ~80 nm, similar to the size obtained from the finite element simulations of the electrostatic properties. These values are also compatible with the electronic inhomogeneities typically observed in LAO/STO, which correlate with structural effects[15,40,41].

The island couples to superconducting reservoirs, which can be inferred from the voltage gap $V_{gap} \approx \pm 30$ μV that separates the CDs in positive and negative bias direction[42,43]. As expected, $V_{gap}$ vanishes when a perpendicular magnetic field $B = 1$ T is applied (see Supplementary Note 4). We further observe pronounced negative differential conductance (NDC) along the edges of the CDs, which can be related to the sharp changes in density of states in the superconducting reservoirs around $\pm\Delta$. Since NDC occurs symmetrically for both positive and negative bias, we conclude that both reservoirs exhibit a superconducting energy gap (see Supplementary Note 5). When we compare the value of $V_{gap} = 2\Delta$ with the superconducting gap in the reservoirs, $\Delta \approx 22$ μeV, we obtain reasonable agreement. We note that in this regime IV the level spacing of quantum states on the island is of the same order as the superconducting gap, $\delta\varepsilon \sim \Delta$. We are therefore in the limit of Anderson's criterion of superconductivity at small scales ($\delta\varepsilon < \Delta$)[44,45].

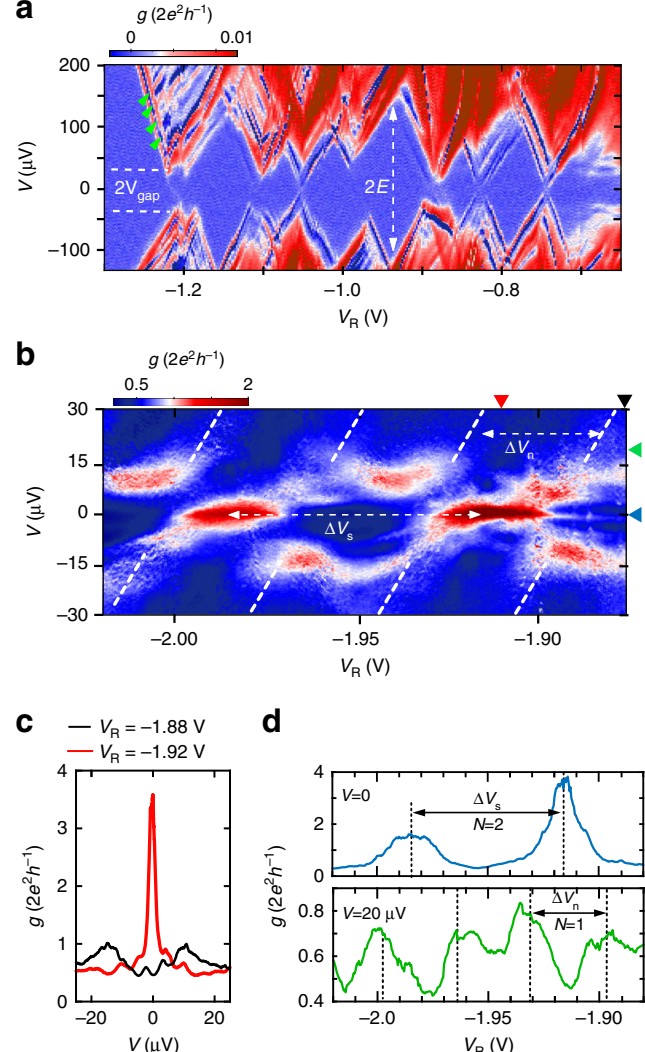

Fig. 4 Charge island in regimes IV and III. **a** Conductance diamonds measured in regime IV with voltage bias $V$ on the vertical axis. $E$ denotes the addition energy of the island. The diamonds exhibit a gap of $V_{gap} \approx \pm 30$ µV. Green arrows indicate signatures of excited states due to quantum confinement. Note that due to gate history effects, this regime occurs at a lower $V_R$ range. **b** $g$ measured in regime III. For small voltages $g$ exhibits peaks with periodicity $\Delta V_s = 70$ mV. For $V > 15$ µV the periodicity increases by a factor 2, $\Delta V_n = 35$ mV (dashed white lines). Black, red triangles and blue, green triangles indicate the line cuts shown in **c**, **d**, respectively. **c** vertical line cuts from **b** at $V_R = -1.88$ V and $V_R = -1.92$ V. **d** Horizontal line cuts from **b** at $V = 0$ (top panel) and $V = 20$ µV (bottom panel). The change in periodicity by a factor 2 suggests a change in number of transferred charges from $N = 2$ to 1, indicative for a superconducting island

### Charge island with superconducting ground state.

Finally we turn to the strong coupling regime III [Fig. 4b]. The pattern of gapped CDs is not visible here. Instead we observe zero bias conductance peaks which are of the order of the quantum of conductance, $g \geq (2e^2h^{-1})$, [cf. Fig. 4c, red curve]. They alternate with regions where $g$ is suppressed. This suggests that the island is more transparent in this regime, allowing for Cooper pair transport[46] at zero bias. The peaks in $g$ occur periodically in $\Delta V_R$, with a periodicity $\Delta V_s = 70$ mV [Fig. 4d, top panel]. Above a certain bias voltage $V \approx \pm 15$ µV, the periodicity changes by a factor 2, $\Delta V_n = 35$ mV [Fig. 4d, bottom panel]. This suggests that the parity of the island influences its energy state, as expected for a superconducting island[42]. In its ground state the island hosts

Cooper pairs (even parity) and thus exhibits a charging energy $2U$, reflecting the Cooper pair's charge $2e$ ($N = 2$). Above a critical bias voltage the odd-parity state becomes available for quasi particles in the reservoirs thus enabling single electron transport across the island ($N = 1$). This results in period doubling of the Coulomb blockade oscillations. Our data therefore suggest that in the strong coupling regime III the island is in a superconducting state, thus forming a superconducting quantum dot.

## Discussion

We have realized a SQPC with split gate technology in a 2D superfluid. Because the superconducting point contact and the superconducting equilibrium reservoirs are made from one and the same material, transport becomes independent of unknown material interfaces, different Fermi velocities and atomic mismatch. Our system can serve as a unique experimental platform for future experiments on 2D superconductivity. This will make it possible to study, for example, the microscopic properties of the LAO/STO interface superconductivity, but also the properties of genuinely SQPCs as originally envisioned[3]. It may furthermore enable the investigation of nano scale superconductivity in few electron quantum dots.

## Methods

**Device fabrication.** We use single crystal $TiO_2$ terminated, (001) oriented $SrTiO_3$ (Crystec ©GmBH) as a substrate without further modification. The fabrication involves three electron beam lithography steps (EBL), which are carried out using a double layer resist (PMMA 495K/950K, thickness 100/200 nm, baked for 15 min at 175 °C) which is exposed with a dose of typically 800–900 µCcm$^{-2}$ and developed using a MIBK:IPA, 1:3 solution (90 s). The first EBL step defines the positions of reference markers which are obtained by sputtering 60 nm Tungsten ($W$) at pressure $p = 0.02$ mbar and consecutive ultrasonic lift-off. The second EBL step patterns the geometry of the device: Those regions which are to remain insulating are covered with 20 nm of sputtered $AlO_2$ ($p = 0.003$ mbar, 200 W power, 20 sccm Ar flow; lift-off process in 50 °C acetone). Next, the LAO layer is grown by means of pulsed laser deposition (PLD) at 770 °C with an $O_2$ pressure of $p_{O2} = 6 \times 10^{-5}$ mbar. Only in those regions which are not covered by the $AlO_2$ hard mask growth is crystalline such that the STO surface is covered with a 12 unit cell (5 nm) LAO layer, giving rise to the 2DES at the interface. In all other regions the $AlO_2$ mask prevents the formation of the 2DES and the LAO layer is amorphous. Growth is monitored in situ by reflection high energy electron diffraction which confirms layer-by-layer growth. After LAO deposition, the sample is annealed for 1 h at 600 °C and at a pressure of $p_{O2} = 300$ mbar in order to suppress the formation of $O_2$ vacancies. The final EBL step defines the pattern of gate electrodes. Polymer residuals are removed with an Oxygen plasma (15 s, 200 W, 212 sccm $O_2$ flow). The metal layer for the surface gates is deposited using electron beam evaporation ($p_{base} \leq 5 \times 10^{-8}$ mbar). This layer consists of 100 nm Au. No sticking layer is used. The first 20 nm Au are deposited at a rate of 0.5 Å s$^{-1}$. Then the rate is increased to 1 Å s$^{-1}$ until the final thickness of 100 nm is reached. Using a syringe for the lift-off with 55 °C acetone ensures a gentle procedure that prevents the gates from peeling off. The sample is then mounted in a chip carrier with silver paint, serving as a back gate. Ultrasonic wedge bonding provides Ohmic contacts to the 2DES.

**Electrical measurement setup and device characterization.** All measurements (unless stated otherwise) are performed using dc electronics, with the current sourced at reservoir S of the sample and drained at reservoir D. The resulting voltage drop $V$ is probed at separate contacts in the respective reservoirs. The dilution refrigerator is equipped with copper powder filters, which are thermalized at the mixing chamber, and Pi-filters at room temperature. The carrier density in the 2DES is adjusted globally by applying a negative back gate voltage $V_{BG} = -1.875$ V, which corresponds to a reduced density compared to $V_{BG} = 0$. We determine the carrier density from Hall measurements performed at 300 mK using voltage probes on opposite sides of the reservoir with width $w = 150$ µm. The longitudinal resistance is determined from voltage measurements between probes separated by l = 112.5 µm. This yields a carrier density $n \approx 3 \times 10^{13}$ cm$^{-2}$ and a mobility $\mu \approx 800$ cm$^2$ V s$^{-1}$. For this carrier density we observe the resistively measured superconducting transition at $T_c \approx 100$ mK, which corresponds to a BCS gap $\Delta_{Tc} = 15$ µeV.

**Data availability.** The data that support the findings of this study are available from the corresponding author upon reasonable request.

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

## Acknowledgements

We thank L.M.K. Vandersypen for valuable comments on our manuscript. This work was supported by The Netherlands Organisation for Scientific Research (NWO/OCW) as part of the Frontiers of Nanoscience program, the Dutch Foundation for Fundamental Research on Matter (FOM) and the European research council (METIQUM, grant no. 339306). The research leading to these results has further received funding from the European Research Council under the European Union's H2020 programme/ERC Grant Agreement n. [677458].We acknowledge received funding from the project Quantox of QuantERA ERA-NET Cofund in Quantum Technologies implemented within the European Union's Horizon 2020 Programme. T.M.K. further acknowledges support from the Russian Science Foundation (RSF) Project No.17-72-30036 .

## Author contributions

S.G., T.M.K., and A.D.C. conceived the experiment. E.M. fabricated the samples. E.M. and H.T. carried out the experiments. H.T. and T.M.K. analyzed the data with input from E.M. N.M. carried out the finite element simulations. H.T., E.M., and T.M.K. wrote the manuscript. All authors commented on the manuscript. A.D.C. supervised the project.

## Additional information

**Competing interests:** The authors declare no competing interests.

