## [Peer Review File · Nature Communications]

Reviewers' comments:

Reviewer #1 (Remarks to the Author):

This is a very nice work where the operation of a superconducting quantum point contact with split gates in the two dimensional LAO/STO superfluid is demonstrated.

The work is excellent from a technological point of view. The device is quite complicate to realize and sophisticated material science and nanofabrication techniques are used to this aim.

In terms of relevance and novelty of the work I have some concerns on the manuscript. As properly pointed out by the Authors, in literature there are other examples of devices with similar performances. Apart from the novelty of the layout with respect to the device using a semiconductor as a barrier, it is not clear whether the devices realized in this work offers better performances. They do not compare the outputs with those of alternative solutions. Does it offer new functions? How many samples have they tested? What about reproducibility and yield?

You could expect that this kind of non-standard device could at least clarify the physics of the employed innovative material. This does not seem to be the case. What kind of novel information can the outputs of the measurements give on the physics of the device and LAO/STO systems?

For these reasons I think that this manuscript is not suitable for publication in Nature Communications. It is not really of wide interest and more appropriate for a technical journal.

I have a minor comment concerning Fig. 2. This kind of plot hides several important information and features of the I-V curves are 'squeezed'. I-V curves should be clearly shown, as partly done in the Supplementary Material, for all the regimes and fitted through classical models on weak links. This is necessary (and should be partly inserted in the main text) to support more firmly the transition from one transport regime to another.

Reviewer #2 (Remarks to the Author):

Thierschmann et al. report a split-gate-defined superconducting weak link in LaAlO₃/SrTiO₃. As they point out, this is novel: other gate tunable superconducting weak links have generally required two separate materials -- the superconducting leads and the semiconducting, tunable link. Their work removes the interface issues and thus offers the promise of cleaner junctions. This is a careful, high-quality study. My main concern is about how sharp the conclusions to be reached are, given that the "quantum point contact" cannot be tuned to have some small number of nearly perfectly-transmitting channels, unlike those in semiconductor heterostructures such as GaAs/AlGaAs. Notably, the authors critique atomic break junctions as mixing mesoscopic quantum channels with atomic/molecular wavefunctions. But I see those studies' main weakness as having multiple partly transmitting channels, requiring a fit to a multiparameter transport model to draw any conclusions. That same situation holds in the present work. Similarly, arXiv:1706.09150 shows rather similar phenomenology to the gate-tunable weak link in the present work, though it is based on an InAs nanowire with superconducting leads (and thus has heterointerfaces through which charge carriers must be transmitted).

Contrast this with arXiv:1705.05049 (again based on InAs, though in this case with a 2DEG and a split gate). That work seems to have simpler and better-defined (integer number of fully-transmitting modes) superconducting QPCs than the present work. It's a different material system, and the present work is clearly important and distinct, but I think it would be worth putting it more solidly in the context of emerging gate-tunable mesoscopic superconductive systems, particularly in InAs with epitaxial Al contacts.

The observation of a Coulomb diamond regime in the present work is interesting and somewhat surprising as the authors note, given the strong dielectric screening in SrTiO₃ at low temperatures. I would like to see data on an intentionally-formed quantum dot in SrTiO₃, which would be more controllable than just some regime of operation of a nominally-single constriction. I realize doing additional experiments is time-consuming. But maybe the authors tried it and could report briefly on the outcome.

Finally, I appreciate an overall outline of the fabrication process, but given the novelty and special challenges of doing 100-nanometer-scale gating in SrTiO₃ I would expect to see far more detail about the process (which resist used, bake temperature, development process, sputter parameters, etc.). Were the Au gates truly deposited with no sticking layer? If so, was there a problem with adhesion?

Overall, before recommending publication in Nature Communications I would like more clarity about what is learned from this work and how it is distinct from recent work on other tunable Josephson Junctions.

Reviewer #3 (Remarks to the Author):

This manuscript discusses experiments involving superconducting LAO/STO 2DEG with split top gates. Different regimes are observed depending on the values of the gating, ranging from quantum dot behavior to full superconducting transmission in the limit of few channels.

I have concerns about the claims of “novelty”. The work presented here bears a strong resemblance to results published recently in Nature Comm by Prawiroatmodjo et al (doi:10.1038/s41467-017-00495-7). They too uses the split gate geometry in LAO/STO, and show many strikingly similar results.

I am suspicious of the role that the thin gate shown in the actual device (Fig 1c) may or may not have on the the transport. The caption states that it is “not used in the experiments”, but according to the diagram it is grounded. How can the authors be sure it plays no role in the device properties? It is maybe 200 nm from the left and right split gates. A related concern: judging from the geometry of the split gates, it is not clear how there could be a real confining potential that is determined by the wedge shape as opposed to some irregularity at the apexes of the gates. There only seems to be a single device. How reproducible is this work?

Looking at the simulations, I have concerns there as well. Figure 5 in the supplement shows a geometry in which the lateral dimensions of a dot with diameter “d” are artificially introduced. No such circular confinement exists in their actual devices. The left and right boundary conditions that assume a constant V_{source} and GND are also quite artificial. It is certainly a simulation of something, but it is not clear that the simulation bears any resemblance to the actual geometry. Where is the confinement in the quantum dot coming from? There is nothing really wrong with studying an accidental QD, but that cuts into the main claim that the properties of these devices are being deterministically engineered.

I don't think that this paper rises to the level of novelty or significance required for publication in Nature Communications.

Here are some more comments:

The number of channels is hard to determine from the $g(V_r)$ plot in Fig 3(d). How do we know for sure that there are few channels with high probability of transmission, as opposed to many channels with low probability of transmission?

There looks to be an error in the labeling of the V_R scale in Fig 4(d). The sequence reads -2.0, -1.95, -0.90.

Reviewer #1 (Remarks to the Author)

This is a very nice work where the operation of a superconducting quantum point contact with split gates in the two dimensional LAO/STO superfluid is demonstrated.

The work is excellent from a technological point of view. The device is quite complicate to realize and sophisticated material science and nanofabrication techniques are used to this aim.

Our reply: We thank the reviewer for a careful review of our manuscript. We appreciate the above comment and like to emphasize that it creates the basis for a good experimental study of the properties of superconducting quantum point contacts.

1. *In terms of relevance and novelty of the work I have some concerns on the manuscript. As properly pointed out by the Authors, in literature there are other examples of devices with similar performances.*

Our reply: Before we go into the details below we like to make a comment about the sentence ‘in literature there are other examples of devices with similar performances’. It is not entirely clear what literature the reviewer is referring to (there are no examples provided). Additionally, our focus is not on practical performance but rather on creating a well-defined experimental configuration, which has the potential to reveal the underlying physics. We are aware of several older and more recent experiments using, what we call hybrid Josephson-junctions, with a gatable semiconductor as a weak link, between two not-gatable superconductors. Our work is different from these predecessors in the sense that we have a non-hybrid system in which the ‘weak link’, the constriction, and the superconductors are made of one and the same material. We have been able to do this because our superconductor is one of the rare examples of gate-tunable superconductivity, because of its 2-dimensionality and its low carrier density. This is the essential novelty of our experiment. From a physics point of view, it means that interfaces, different Fermi-velocities, and atomic mismatches do not play a role. As a consequence, electrons emerging from a superconducting reservoir can pass through the constriction without suffering any influence from changes in materials. The constriction is created purely electrostatically. The situation can be compared to older work with superconducting mechanical break junctions with one significant difference that in our case the Fermi wavelength is much longer than the interatomic distance, which implies that the atomic scale is irrelevant, like in semiconductor quantum point-contacts and quantum dots.

Therefore, we believe that we can faithfully argue that our system is different compared to all systems studied before.

2. *Apart from the novelty of the layout with respect to the device using a semiconductor as a barrier, it is not clear whether the devices realized in this work offers better performances. They do not compare the outputs with those of alternative solutions. Does it offer new functions?*

Our reply: As has been made clear our aim is not improved performance but improved physical conditions and exploiting new physical possibilities. For instance, our technology enables now the fabrication of tunable superconducting quantum dots, which can be probed with superconducting point contacts.

3. *How many samples have they tested? What about reproducibility and yield?*

Our reply: We have studied several samples which all showed similar behavior. Quantum dot behaviour was observed frequently and in devices with various gate layouts, including Coulomb blockade diamonds, a transport gap at small voltage, signatures of excited states and a transition to a regime similar to the one that we call regime (ii) in the main text.

We were able to identify a superconducting few-mode QPC in at least one of the other devices.

We include the data from this device in the supplementary material (section 9).

4. *You could expect that this kind of non-standard device could at least clarify the physics of the employed innovative material. This does not seem to be the case. What kind of novel information can the outputs of the measurements give on the physics of the device and LAO/STO systems?*

Our reply: This is a very good question, which has of course been driving our research. Unfortunately, the experimental work goes step by step, and at the present stage, we have conclusively proven that we have achieved the regime where we think we should be. As usual, a lot of physics is entangled. We have a 2D superconductor with a relatively low energy gap, we also have a tunable constriction, and we have a system that evolved easily into puddles, as revealed by the Coulomb blockade. Finally, we have a supercurrent which is easily disrupted by phase-slip events as is evident from the rounding of the I-V curves. The latter is usually cast into the framework of macroscopic variables, but in our case it can be connected to microscopic variables like the occupation of Andreev bound states created out of a 2D superconductor. It is fair to say that many questions are still to be answered, but all of them are very interesting and new.

For these reasons I think that this manuscript is not suitable for publication in Nature Communications. It is not really of wide interest and more appropriate for a technical journal.

Our reply: We respectfully disagree with the reviewer. We believe that this experimental system should not be perceived as of interest to a technical journal. It is just the opposite, it is most interesting from a fundamental point of view because of all the potential that the system provides. We have now demonstrated that we have reached that regime. We are therefore convinced that our results are not appropriate for a purely technical journal but should be communicated to a broad readership with interest in the various topics mentioned above.

- 5. I have a minor comment concerning Fig. 2. This kind of plot hides several important information and features of the I-V curves are ‘squeezed’. I-V curves should be clearly shown, as partly done in the Supplementary Material, for all the regimes and fitted through classical models on weak links. This is necessary (and should be partly inserted in the main text) to support more firmly the transition from one transport regime to another.*

Our reply: To provide more support for the transitions from one regime to another, as requested by the reviewer, we added a panel (Fig 1 (d)), showing representative I-V curves for each regime. Furthermore, we include extensive I-V data for each of the regimes in the supplementary material (section 3).

We are not sure, however, what ‘*fitting to classical models on weak links*’ would mean. It is quite common to compare data with the so-called RSJ-model. However, this engineering model does not have physical content, apart from the macroscopic phase difference between the two superconductors involved. The interesting point of a realistic system is the energy-dependence (or voltage-dependence), which is very clear in the analysis of the superconducting mechanical break junctions, including the multiple-Andreev reflections depending on the transmissivity of the various modes. In our case it is premature to apply that analysis, which has to be modified for a 2-dimensional system.

Reviewer #2 (Remarks to the Author):

Thierschmann et al. report a split-gate-defined superconducting weak link in LaAlO₃/SrTiO₃. As they point out, this is novel: other gate tunable superconducting weak links have generally required two separate materials -- the superconducting leads and the semiconducting, tunable link. Their work removes the interface issues and thus offers the promise of cleaner junctions.

This is a careful, high-quality study.

Our reply: We thank Reviewer #2 for a careful revision of our manuscript. We appreciate that the reviewer recognizes the novelty and quality of our approach.

1.

- a. *My main concern is about how sharp the conclusions to be reached are, given that the "quantum point contact" cannot be tuned to have some small number of nearly perfectly-transmitting channels, unlike those in semiconductor heterostructures such as GaAs/AlGaAs.*

Our reply: In contrast to the reviewer's statement '*cannot be tuned to have a small number of ... channels*', it is our claim that we have reached that regime. This is in agreement with the measured values of R_n . At these values of R_n and for the given value of the energy gap of this superconductor the expected supercurrent is very small which renders the Josephson coupling energy small compared to kT . Therefore, we do not observe a full zero-voltage supercurrent, because of the presence of thermally assisted phase-slips. Features in the I-V curve reminiscent of the critical current are compatible with values expected for a single channel and τ extracted from the measurements.

Nevertheless, we would like to expand this research further to LAO/STO heterostructures with a higher mobility and a smaller tendency to create 'puddles' at lower carrier density. Ballistic transport has been shown over several 100 nm in LAO/STO (Tomczyk et al, PRL 117, 096801 (2016)). This means that a SQPC with perfectly transmitting channels is within reach. The present finite transmission coefficient is presumably a result of scattering events inside the weak link.

- b. *Notably, the authors critique atomic break junctions as mixing mesoscopic quantum channels with atomic/molecular wavefunctions. But I see those studies' main weakness as having multiple partly transmitting channels, requiring a fit to a multiparameter transport model to draw any conclusions. That same situation holds in the present work.*

Our reply: This is an important point in our discussion. In essence we are very positive about the atomic scale mechanical break junctions, in comparison with the semiconductor-based hybrid junctions, with the uncontrolled interfaces. In fact, these junctions have been very successfully compared with the theory by using a large range of experimental curves for different temperatures, which definitely allowed a very accurate set of transmission channels with their transmissivities (the PIN-code of the contacts). In our case, we try to achieve the same for a 2-dimensional superconductor. In addition, we point out that in this system as well as in the standard GaAs/AlGaAs point contacts we should be immune for the atomic scale because the Fermi-wavelength, which will control the number of modes is much longer than the interatomic distance. This is definitely not meant as a critique of the atomic scale work, but meant as an additional observation for our system, which needs to be taken into account in comparing the two. Of course, we believe that our system has extra potential such as the possibility to create and study superconducting quantum dots.

2. *Similarly, arXiv:1706.09150 shows rather similar phenomenology to the gate-tunable weak link in the present work, though it is based on an InAs nanowire with superconducting leads (and thus has heterointerfaces through which charge carriers must be transmitted).*

Our reply: We are aware of this interesting manuscript of Goffman et al, published in New J. Phys. 19,092002 (2017). It reports on InAs wires proximitized by an epitaxially grown Al shell. In a region of approx. 200 nm the shell is removed and two side gates are patterned close to the wire. V-I curves are taken at low and high T at different gate voltages for which G changes from 2.5 to $<0.5 (2e^2/h)$. The MAR-fingerprint-technique (PIN-code) is applied to the VI-curves. Best fits are obtained for 4 channels with different τ . As the authors make clear in comparison to the atomic scale junctions the results are satisfactory, but at the same time leave many problems unresolved, which are partially related to the hybrid nature and to the sensitivity of the semiconductor to impurity charges and doping etc. The τ are varying considerably and highly non-monotonous with gate voltage. This occurs even when the conductance remains almost constant.

We like to add that in our case the tunability is focused on the confinement potential for the superconductor, whereas in this experiment it is the carrier density in the semiconductor, which is addressed.

Contrast this with arXiv:1705.05049 (again based on InAs, though in this case with a 2DEG and a split gate). That work seems to have simpler and better-defined (integer

number of fully-transmitting modes) superconducting QPCs than the present work. It's a different material system, and the present work is clearly important and distinct, but I think it would be worth putting it more solidly in the context of emerging gate-tunable mesoscopic superconductive systems, particularly in InAs with epitaxial Al contacts.

Our reply: We are also aware of this work, with partially the same authors. It is an example of excellent research on hybrid systems. Their work is motivated by the search for Majorana-physics with the possible use in quantum computation and therefore take a material with a significant spin-orbit interaction. From that perspective, it takes the hybrid nature of the Josephson-physics as an unavoidable necessity. It nevertheless has to deal with the fact that the physics is dependent on the interfaces and the induced nature of the superconductivity in the semiconductor entangled with the few-mode nature of the transport.

The crucial difference with our work is that we are creating few-mode conduction channels in a constriction of the superconductor itself. The interesting starting point in our system is the gate-tunability of the superconductor rather than the gate-tunability of a semiconductor weak link. In that sense the difference is very clear, although there are clearly also similarities.

- 3. The observation of a Coulomb diamond regime in the present work is interesting and somewhat surprising as the authors note, given the strong dielectric screening in SrTiO₃ at low temperatures. I would like to see data on an intentionally-formed quantum dot in SrTiO₃, which would be more controllable than just some regime of operation of a nominally-single constriction. I realize doing additional experiments is time-consuming. But maybe the authors tried it and could report briefly on the outcome.*

Our reply: We share the excitement of the reviewer about such experiments and we do plan to address these questions in the future. Unfortunately, at this stage we have no such data available. Concerning the present work, however, we think that the current set of data is sufficient.

- 4. Finally, I appreciate an overall outline of the fabrication process, but given the novelty and special challenges of doing 100-nanometer-scale gating in SrTiO₃ I would expect to see far more detail about the process (which resist used, bake temperature, development process, sputter parameters, etc.). Were the Au gates truly deposited with no sticking layer? If so, was there a problem with adhesion?*

Our reply: We have included more details on the fabrication process in the appropriate methods section.

We note here briefly that indeed no sticking layer was used for the Au gates. To avoid problems with adhesion we were careful to use a syringe to ensure a gentle lift-off process.

Overall, before recommending publication in Nature Communications I would like more clarity about what is learned from this work and how it is distinct from recent work on other tunable Josephson Junctions.

Our reply: We hope we have clarified in the above the contrast and the similarities. The essence is that we deal with a tunable superconductor and not with a tunable junction between untunable superconductors

Reviewer #3 (Remarks to the Author):

This manuscript discusses experiments involving superconducting LAO/STO 2DEG with split top gates. Different regimes are observed depending on the values of the gating, ranging from quantum dot behavior to full superconducting transmission in the limit of few channels.

- 1. I have concerns about the claims of “novelty”. The work presented here bears a strong resemblance to results published recently in Nature Comm by Prawiroatmodjo et al (doi:10.1038/s41467-017-00495-7). They too uses the split gate geometry in LAO/STO, and show many strikingly similar results.*

Our reply: We like to thank the reviewer for his/her critical comments on our manuscript.

Our work focusses on the formation of a superconducting quantum point contact. It closes an important gap in the range of experimental platforms available to perform well controlled experiments in the few mode quantum transport regime of a superconductor with equilibrium reservoirs. Being carried out in a single superconducting system, our experiments rigorously remove the need to either ignore or inconveniently include existing material interfaces into the theoretical models, as it is the case for the widely used superconductor/semiconductor hybrids.

The problem articulated and addressed by Prawiroatmodjo et al. is very different. It focuses on the electronic properties of an emerging charge puddle, which is analysed with the focus on on-site electron-electron interaction. The formation of a superconducting QPC is not addressed. Therefore, the problem addressed in our work and the results obtained are clearly distinct from those of Prawiroatmodjo et al. We were unaware of their work at the time of submission. Since it has appeared now we have decided to include a reference to their work in the context of the Coulomb blockade regime of our data.

- 2. I am suspicious of the role that the thin gate shown in the actual device (Fig 1c) may or may not have on the the transport.*

Our reply: We have also carried out experiments on a device which did not contain a thin gate but only the two wedge shaped gates. Here we identified a few mode SQPC and we observed the formation of a QD. We have therefore empirically ruled out that the thin gate plays a major role for the outcome of our experiments, which we also expect (see below). The set of data on this

other device was less complete and therefore we have focused on a presentation of a more complete set of the other device.

We have included these data in the supplementary material (section 11) and we added a note about the reproducibility of our results explicitly in the main text (at the beginning of the second paragraph).

- a. *The caption states that it is “not used in the experiments”, but according to the diagram it is grounded.*

Our reply: In order not to use it in the experiments, the thin gate was kept at ground potential to ensure that it does not influence the measurements. This a well-established technique for top-gate defined structures, which is frequently applied also in the field of semiconductor devices.

If the gate is not put on any well-defined potential, i.e. if it is kept floating, its actual potential would be un-controlled. Hence, its influence on the 2DES would be unknown.

However, if we set the gate to ground potential its influence on the 2DES is very well controllable: its voltage with respect to the 2DES cannot become larger than the bias voltage applied between the source/drain contacts, which is in our case below 1mV. It can therefore safely be neglected given that the required gate voltages to electrostatically modify the 2DES are orders of magnitude larger ($\sim V$).

To avoid confusion for the reader we have clarified this in the caption of Fig.1.

- b. *How can the authors be sure it plays no role in the device properties? It is maybe 200 nm from the left and right split gates.*

Our reply: See the arguments given above on question 2 and 2a.

3. *A related concern: judging from the geometry of the split gates, it is not clear how there could be a real confining potential that is determined by the wedge shape as opposed to some irregularity at the apexes of the gates.*

Our reply: It is not entirely clear to us what the reviewer is referring to exactly. Let us express our view by referring to the common practice for GaAs/AlGaAs heterostructures.

The wedge shape of the split gates was used very successfully in the original work by van Wees et al in 1988 on quantum point contacts in semiconductors and since then it has become the well-established split gate geometry for semiconductor QPCs. Extensive studies have been published, which use the Poisson-equation and the geometry and include the fact that the plane of the 2-dimensional electron gas (2D superconductor) is at a certain distance of the plane where the gates are. It also takes into account the dielectric constant of the material.

The roughness of the gate edges caused by the Au grains is not expected to affect the shape of the confining potential significantly because irregularities in the electric field become smeared out with increasing distance from the gate.

We note that the image shown in Fig. 1b) was taken using an Atomic Force Microscope. With this technique small particles which may be picked up by the cantilever tip during scanning, can make the size of small irregularities appear strongly enhanced because the generated image is a result of a convolution of the actual tip shape with the sample topography.

A Scanning Electron Microscope image of a different gate structure is presented below which shows the clean edges of the Au gates obtained with our technology and which was typical for the device used for the measurements.

To reassure the reviewer further, we like to point out that the image shown in Fig. 1b) was not taken from the very same device that was used in the measurements but from another device with the same layout. The reason is that the original device has been damaged due to electrostatic discharge.

We like to add also, that there is a difference regarding the confining potential for GaAs/AlGaAs heterostructures and those for 2-dimensional superconductors. For the latter the depth of the confining potential is less deep than for the semiconductor. This has an effect on the tendency to form an inhomogeneous landscape in-line with the interpretation provided by Prawiroatmodjo et al. We expect to be able to shed more light on these differences in future research.

4. *There only seems to be a single device. How reproducible is this work?*

Our reply: See above (question 2.)

5. *Looking at the simulations, I have concerns there as well. Figure 5 in the supplement shows a geometry in which the lateral dimensions of a dot with diameter “d” are artificially introduced. No such circular confinement exists in their actual devices.*

Our reply: The simulations are carried out to investigate the influence of the gate voltages on the charging energies of a charge puddle between the gates. We do not aim to reveal with them how the dot is being formed. Finite element simulations are a precious but limited tool to perform a consistency check on an interpretation of experimental results. In our case the presence of a charge puddle is evident from the experimental results. Its presence is then introduced into the model. Obviously, since the exact shape of the dot is unknown, the assumed circular shape is an approximation.

The simulations show that the voltage applied to the top-gates locally reduces the dielectric number of the STO substrate. They confirm that this has a strong effect on the charging energy of a charge puddle formed in the 2DES region between the gate tips. As we describe in the main text, the Coulomb charging energy therefore should be expected to become relevant even for puddles much larger than only a few nanometers.

If we assume a circular shaped puddle as a first approximation, we obtain an estimated puddle size which is consistent with the spectrum of excited states inferred from the transport spectroscopy data. This is the essential conclusion that we draw from these simulations, as explained in the main text.

Because the exact shape of the puddle is not known, as the reviewer points out correctly, the results of the simulations only allow for an order of magnitude estimate, which is what we do. Therefore, this simplifying assumption is justified for our purpose,.

To further highlight the relevance of taking into account the unusual dielectric properties coming from the STO substrate, we have extended the supplementary information by calculations of the charging energy if effect is neglected. The results show that the resulting charging energy differs by one order of magnitude.

6. *The left and right boundary conditions that assume a constant V_{source} and GND are also quite artificial.*

Our reply: The assumption of having one reservoir on GND and the other reservoir at a constant bias potential V_{source} reflects the experimental conditions of a voltage-biased measurement accurately (as stated in the main text at the beginning of the discussion of regime (iii)).

7. *It is certainly a simulation of something, but it is not clear that the simulation bears any resemblance to the actual geometry.*

Our reply: We disagree with this statement. The simulation clearly bears resemblance with the actual geometry of the device:

The geometry of the gates, the spatial extensions of the LAO/STO interface and the layer stack of STO/LAO and STO/LAO /Au are chosen precisely as in the experiment. The model includes the electric field and temperature dependence of the dielectric properties of the system. The electronic configuration of gate voltages and source/drain voltages are chosen as in the experiment. Based on the experimental results, it is a highly plausible assumption that the charge puddle is formed in the region of the 2DES between the tips of the split gates.

The simulation therefore realistically captures the dielectric environment of a charge puddle which is situated between the gates. It therefore allows for an order of magnitude estimate of the charging energy expected for certain puddle dimensions. This is what its output is used for in the context of this manuscript.

8. *Where is the confinement in the quantum dot coming from?*

Our reply: The reviewer raises a very interesting question. The origin of the confining potential which gives rise to the quantum dot in our device is not entirely clear. As we point out in the first paragraph of the discussion of regime (iii), one possible explanation could be drawn from analogies to semiconductor devices. There disorder can induce inhomogeneities in the electrostatic potential landscape which sometimes leads to a confinement potential and, thus to quantum dot behaviour, in 1D systems close to charge carrier depletion. A similar explanation is proposed by Prawiroatmodjo *et al.*

However, in order to provide a more conclusive answer to this question one would presumably have to combine transport experiments with other experimental techniques such as scanning probe experiments. Without such experimental data available, however, any further discussion of this question in our manuscript would remain speculative.

We note, however, that it will indeed be very interesting to address in future experiments in greater detail the mechanism that drives the formation of the quantum dot and especially to study how the charge puddle formation relates to current theories on the superconductor/insulator transition in LAO/STO.

9. *There is nothing really wrong with studying an accidental QD, but that cuts into the main claim that the properties of these devices are being deterministically engineered.*

Our reply: The wording of the reviewer ‘deterministically engineered’ are somewhat surprising, because it is a claim that we would love to make, but we are realistic enough to recognize that full control has not been achieved (if that ever exists). What we have tried to make clear that we have been able for the first time to create a few-channel superconducting quantum point contact by using one and the same material for the constriction and the superconducting reservoirs. This is an accomplishment from which we can further build as we intend to do.

I don't think that this paper rises to the level of novelty or significance required for publication in Nature Communications.

Our reply: As we have outlined in our replies above, we believe that our work is very distinct from other work. We demonstrate that the superconducting LAO/STO interface 2DES provides a unique experimental platform to address problems on the Josephson effect in the regime of few mode quantum transport with potential for a number of interesting follow-up problems.

Here are some more comments:

10. The number of channels is hard to determine from the $g(V_r)$ plot in Fig 3(d). How do we know for sure that there are few channels with high probability of transmission, as opposed to many channels with low probability of transmission?

Our reply: We like to point out that this information is provided in the main text. We calculate τ through Z which is extracted from the measured excess current. Using this transmission for a ballistic few channel QPC gives the right critical current.

If it was many channels with low transmissivity we would get a much lower critical current as indicated by the blue dashed line in the bottom panel in Fig. 3(e).

There looks to be an error in the labeling of the V_R scale in Fig 4(d). The sequence reads -2.0, -1.95, -0.90.

Our reply: We thank the referee for pointing out this error.

Reviewers' comments:

Reviewer #1 (Remarks to the Author):

In the revised manuscript and in the rebuttal letter, the Authors did not give any additional argument on the innovation of their work, just repeating arguments which were clear also in the original manuscript.

I do not understand why the Authors reported only a part of my previous sentence:

“As properly pointed out by the Authors, in literature there are other examples of devices with similar performances.”

I obviously refer to the fact that this is not the first superconducting quantum point contact; and their device does not seem to demonstrate to date any novel functionality. Some of the references are given by the same Authors even in their abstract. It would have been different if the Authors would have used the properties of a superconducting quantum point contact to reveal new physics of the LAO/STO system. But this does not seem to be the case either. The degree of novelty is therefore incremental and not transformative, and not enough in my opinion for a paper for Nat. Comm.

Concerning the minor mandatory point, the Authors ask for clarifications, that I will be happy to provide though they are quite obvious.

When the Authors in fig. 3e top panel pretend to give I_{ex} without any error bar and on the scale of 0.1 nA, I think that they need to have a modeling in mind, that has to be also quite accurate. The current is an excess current compared to what!! I understand that differently from other works on quantum point contacts (ref.4 of their list), it is difficult to model energy dependent processes in this configuration, but a rough comparison with engineering models would be the minimum for a reader to “roughly” understand the general properties of the junctions and to better appreciate the exotic features of the junctions. There are several issues on the shape of the I-V curves that should be briefly addressed to justify the discussion of the Authors.

Reviewer #2 (Remarks to the Author):

I have read the rebuttal letter and the changes made. I feel the authors have thoroughly and effectively responded to not only my comments and questions but also those of the other referees. I particularly appreciate the addition of a substantial dataset from a second device. Despite substantial scientific questions remaining (to be explored in future work) I feel the modeling is very reasonable, in contrast to what Referee 3 expresses.

I still think it would be useful for the community (and appropriate in a paper like this) to document the fabrication processes in much more detail than is done in Supplementary Info section 1, though I appreciate the overview that's been added. When working with materials other than conventional metals and semiconductors, every detail, from resist used, to bake temperature, to developer, to beam energy (and on, and on) can be important. Even though the researchers may not feel they have carefully optimized the process, they do have a working process, and this should be shared as a starting point for others' work to corroborate and extend this work.

With such information added, I strongly support publication in Nature Communications.

Reviewer #3 (Remarks to the Author):

I have read the revised manuscript, and the referee response. Some of the issues are addressed but not all of them.

I withdraw my complaint about novelty with respect to the work of Prawiroatmodjo et al, which was published only recently. It is properly referenced in the revised manuscript. The experimental work is strikingly similar even though the focus is "different". The present authors also appear to be working with a "charge puddle" of unspecified origin.

It is nice to see that there is a second device now in the supplement. But there is very little discussion of the data, and there are many qualitative differences between the data that is well described in the main text and the different device in the supplement. The authors describe four regimes: (iii), (ii), (i), (o). I would say that regime (iii) looks similar in the second device, but the resistance plot (Fig.2) in regime (ii) looks quite different from Fig. 12(ii). Even (iii) does not resemble the main text but there is some indication of coulomb diamonds forming. It's not clear why the other regimes were not shown. Also, the diamonds are offset in Fig. 12. The authors should comment on this feature. The second device seems to reach a zero-resistance state, while the device in the main

text looks to have a ~ 1 kOhm background resistance. That difference should also be commented upon.

The authors place great importance on having created a split gate geometry. The very first sentence of the abstract reads:

"One of the hallmark experiments of quantum transport is the observation of the quantized resistance in a point contact formed with split gates in GaAs/AlGaAs heterostructures."

However, the authors have not reached this "hallmark". The authors provide no data that would support the existence of QPC behavior as first demonstrated by van Wees back in 1988. Where are the conductance steps? Instead of observing the characteristic features of a QPC, the behavior seems to be more related to a quantum dot that is not part of their design. The physics of a few mode superconducting channel is interesting but there have been other reports of narrow superconducting devices, although not with the split gate. It is not clear how the split gate is acting on the system or how it is modifying transmission through the device.

The main issue that I have still is the impression that the manuscript gives that the split gates are somehow directly responsible for the behavior. There is no geometric feature in either device gate that can be said to produce a quantum dot, and yet much of the data seems to be defined by this quantum dot behavior. The simulation that has a rather large mysterious quantum dot in between the gates. And not evidence of ordinary van Wees QPC behavior is provided.

In summary, my concern is that the paper as written is highly misleading. I cannot recommend publication in this journal.

Reviewer #1 (Remarks to the Author):

In the revised manuscript and in the rebuttal letter, the Authors did not give any additional argument on the innovation of their work, just repeating arguments which were clear also in the original manuscript.

Our reply: We thank the reviewer for acknowledging our response and the changes we have made. We regret that we have not yet been able to convince the reviewer in which sense our work is innovative. This is obviously a failure on our side. And this causes us great concern because we strongly believe, that our work is very different from previous work.

I do not understand why the Authors reported only a part of my previous sentence: "As properly pointed out by the Authors, in literature there are other examples of devices with similar performances."

I obviously refer to the fact that this is not the first superconducting quantum point contact; and their device does not seem to demonstrate to date any novel functionality. Some of the references are given by the same Authors even in their abstract.

Our reply: The reviewer makes clear with the sentence "I do not understand why" that he/she believes we are selectively reading his previous comments. In this context he/she states that "this is not the first superconducting point contact" and neither is there "any novel functionality".

Our responses to these 2 separate points are as follows:

1; Superconducting quantum point contacts and the role of the superconducting state.

We believe we are fully aware of the evolution of superconducting point contacts, which we acknowledge in our references. As the title makes clear, our focus is on the realization of such a SQPC in LAO/STO and its properties. The material we work on, the superconducting state in this material and its tunability, makes all the difference. Therefore, we are convinced that our work is new.

Why is the fact that we work on this topic in LAO/STO so significantly different?

Comparison to previous work:

a; The use of hybrid superconductor-semiconductor systems

Since the early 90s there have been a large number of experiments in which a ballistic semiconductor was contacted by two conventional superconductors. The new element was the ballistic semiconductor, usually a heterostructure, followed later by materials like graphene and semiconductor nanowires. This work had two significant ingredients compared to our work: 1; the electrostatic field effect was acting on the non-superconducting part, the weak link, and not on the superconductor. 2; the interface between the superconductor and the non-superconducting part plays a critical role.

What is very important though, is the *insignificant* role of the properties of the superconductor in these previous experiments. The superconductor contributes only to the experiment, in the sense that it provides a superconducting density of states,

with an energy gap, as well as a macroscopic quantum phase. Otherwise none of the properties of the superconductor are relevant. They are just boundary conditions.

This has very interesting implications for phase-coherent transport, because it is at the heart of Beenakker's reformulation of normal quantum transport into a superconducting quantum point contact. In the limits used in his analysis, the main argument is that the normal state scattering matrix is also decisive for systems in which this scattering matrix is coupled to boundary conditions representing a superconductor with an energy gap and a phase. This is a very interesting result, generalizing the original discovery of Josephson for tunnel junctions, but as such it does not provide new information about the *superconductor*.

The fact that we work with a *non-hybrid system* is therefore a very important distinguishing feature, because the *gatable superconductor itself has now become part of the quantum transport problem*. And the unique aspects of the interfacial superconductor, LAO/STO, is that we can change its carrier-density.

b; Superconducting mechanical-break junctions (MBJ's)

The other group of alternative systems we considered in composing our manuscript are the superconducting mechanical break junctions. We refer to those papers, such as Ref.4, by Scheer et al., to make clear in what way our experiment is significantly different. The elegant part of this group of experiments is that under very clean conditions the superconducting wire, again made of a conventional superconductor, is broken. The broken pieces are then brought back together with atomic scale precision creating a vacuum tunnelling gap between the outermost atoms of the superconductor. This mechanical tunability, in combination with the cleanliness of the *in situ* process, makes it possible to study the energy-dependence of quantum transport of these conventional, 3-dimensional (!), superconductors in beautiful detail. Note that these systems are also non-hybrid.

The difference with our work is; 1; because in LAO/STO the carrier-density is low and tunable, we can tune our superconducting state from superconducting to insulating. 2; like in semiconducting heterostructures, the Fermi wavelength is much larger than the interatomic distance, allowing electrostatic control of the constriction and 3; our superconducting state is 2-dimensional.

The fact that *we work with LAO/STO*, which is an *electrostatically tunable 2-dimensional superconductor*, makes this work very significantly different from the work on mechanical break junctions

2; Novel functionality

Our goal is to perform an experiment, i.e. bring our system to reveal hidden unknown information on a non-conventional superconductor. For that experiment, we use the tools of present-day clean room technology. Since it resembles an electronic component, it has become customary to call it a device. Strictly speaking a device is an object with a certain aimed for functionality. We use the constructed system as a means to carry out an experiment under unusual circumstances. Our aim is not the investigation of a new device with new functionality, which we also do not claim in the manuscript.

Changes made: We have further highlighted the role of the two dimensional nature of the superconductor in our experiments in the introduction.

It would have been different if the Authors would have used the properties of a superconducting quantum point contact to reveal new physics of the LAO/STO system. But this does not seem to be the case either.

Our reply:

We appreciate the critical comment from the reviewer because it makes clear to us, that we have been unsuccessful in conveying the essential message of our manuscript to the reviewer. The full claim of our manuscript is that our experiment is revealing properties of the LAO/STO system. It is the essence of our Figs. 2 and 4. They communicate what the LAO/STO system shows if studied in the quantum transport regime. It is a very rich outcome, which needs to be studied further but it shows clearly the role played by inhomogeneities in the 2-dimensional superconductor when the carrier density is depleted electrostatically, which show up as superconducting charge puddles. The fact that we have to invoke *charge puddles which are superconducting* is interesting in itself. In our view it is very clearly a new regime to study the physics of the LAO/STO system. It is a unique realisation of quantum transport in a diluted superfluid which offers many details to be worked out.

Perhaps, it is worth pointing out that this regime is reminiscent of the metal-insulator transition in 2-dimensional electron systems. Upon lowering the electron density, the Coulomb interactions between the electrons increases, ultimately exceeding the kinetic energy as embodied in the Fermi-energy. Strongly correlated systems are complex in itself and experiments are not easily made decisive, but this does not minimize its scientific importance. In the LAO/STO system one has the analogue of the normal state metal-insulator transition, in the sense that it is in this case a gate-tunable superconductor-insulator transition. This is a very important field of study with challenging questions with broad implications (see the recent review by Kapitulnik, Kivelson and Spivak, arxiv:1712.07215). Our experiments make clear that the approach to the insulating state is going through a phase in which the system becomes inhomogeneous and we can unravel the parameters of this inhomogeneous phase, which we intend to do in future work.

The degree of novelty is therefore incremental and not transformative, and not enough in my opinion for a paper for Nat. Comm.

Our reply: We have given above all the arguments that we believe are applicable and we hope that we have convinced the reviewer that our work is indeed novel and distinct. Whether our work is transformative is probably too early to say. We believe that a consistent continuation of our work will have a significant impact on our understanding of the relevant quantum phase transitions.

Concerning the minor mandatory point, the Authors ask for clarifications, that I will be happy to provide though they are quite obvious.

When the Authors in fig. 3e top panel pretend to give I_{ex} without any error bar and on the scale of 0.1 nA, I think that they need to have a modeling in mind, that has to be also quite accurate. The current is an excess current compared to what!!

Our reply: We appreciate this technical remark and we thank the reviewer for pointing out the missing indication for errors in fig. 3.(e), which we have added now. We point out that the errors in this figure are dominated by the uncertainty about the exact value of the superconducting gap. Concerning the excess current, however, in contrast to what the reviewer assumes, there is not a great deal of modelling involved (Blonder et al., PRB 25, 4515 (1982)). Figure 3 makes clear how we have defined the excess current, following a well-established convention in the field of Josephson junctions. Using a linear fit in the high bias regime reduces the error for I_{exc} well below 1%.

The problem for any Josephson-type contact is that, in the presence of any finite voltage, the I,V curves represent time-averaged quantities. Therefore, in most cases there is a tendency to focus on the zero-voltage state. The finite voltage case is to a first order approximation always described by the so-called resistively shunted junction (RSJ-) model. It contains the standard $I_c \sin(\varphi)$ relationship for the Josephson current, with φ being the difference between the macroscopic quantum phase of the two weakly connected superconductors, in parallel to a fully normal Ohmic current: V/R . This engineering model is very well suited to understand the changes in the I,V curve due to the thermal fluctuations causing phase-slips in the Josephson-current and concomitant voltage spikes, which in a time-averaged scale leads to a rounding of the I,V curve, which we deal with in the paper at page 3, left column. For the higher voltages, the Ohmic resistor in the RSJ model leads, unavoidably, to an asymptote which goes through the origin, which is a consequence of the chosen simplified model, known to be incorrect for real systems. Such a resistive shunt is never the real situation for any superconducting weak link (unless one makes on purpose a tunnel junction shunted by a normal resistor). Superconducting weak links always display a voltage-dependence related to the superconducting energy-gap, in particular if the weak link is short compared to the elastic mean free path. In the field of Josephson-junctions the excess current, for voltages above the energy-gap, is then extracted from an extrapolation to the current axis. The straight-line, parallel to a normal state resistance, which intercepts the zero-voltage axis at a certain current value is crucial. The current at this interception is defined as the excess current. This excess current is routinely observed in superconducting contacts and known to reflect the energy-dependence of the density of states in the superconducting electrodes, provided the data are in the regime of voltages higher than the energy gap in the superconducting electrodes. This is the case in our experiment in view of the T_c at the LAO/STO interface.

Changes made: We have refined the analysis (using linear fits for the excess current) presented in in Fig.3(e) and we have indicated the errors in these plots.

I understand that differently from other works on quantum point contacts (ref.4 of their list), it is difficult to model energy dependent processes in this configuration, but a rough comparison with engineering models would be the minimum for a reader to “roughly” understand the general properties of the junctions and to better appreciate the exotic features of the junctions. There are several issues on the shape of the I-V curves that should be briefly addressed to justify the discussion of the Authors.

Our reply: The reviewer is right in the assumption that a full energy-dependent calculation is in this case not reachable. Partly, also because the time-averaged processes have to be taken into account as well, including the effect of thermal noise. The I,V is a mixture of energy-dependent processes and time-dependent processes due to the evolution of the macroscopic phase-difference. Such a one-to-one correspondence can only be obtained in well-defined model-systems, which serve less as an experiment to discover new things but more as a

model-system to obtain numerical accuracy between theory and experiment. This is obviously not our goal in carrying out this experiment. This experiment is meant to discover properties. The excess current, defined for voltages above the energy gap, is taken here as a source of information on the energy dependence, which can be used as such if one goes to voltages above the energy gap of the electrodes. It signals that for each electron in the energy-window of the superconducting electrode the charge transferred is enhanced, to maximally a factor of 2, by the process of Andreev reflection. There is no need to perform detailed modelling to make this argument in analysing the abundance of data in this particular experiment.

Reviewer #2 (Remarks to the Author):

I have read the rebuttal letter and the changes made. I feel the authors have thoroughly and effectively responded to not only my comments and questions but also those of the other referees. I particularly appreciate the addition of a substantial dataset from a second device. Despite substantial scientific questions remaining (to be explored in future work) I feel the modeling is very reasonable, in contrast to what Referee 3 expresses.

I still think it would be useful for the community (and appropriate in a paper like this) to document the fabrication processes in much more detail than is done in Supplementary Info section 1, though I appreciate the overview that's been added. When working with materials other than conventional metals and semiconductors, every detail, from resist used, to bake temperature, to developer, to beam energy (and on, and on) can be important. Even though the researchers may not feel they have carefully optimized the process, they do have a working process, and this should be shared as a starting point for others' work to corroborate and extend this work.

With such information added, I strongly support publication in Nature Communications.

Our reply: We appreciate the compliments made by the reviewer and the support for publication. We sincerely hope that our replies to the other reviewers convinces him/her even more about the significance of our work.

Regarding the request for more details on the fabrication process, we are happy to provide them and have expanded our information regarding this point in the supplementary information.

Changes made: We have added a detailed recipe of the fabrication process to the Supplementary, section I.

Reviewer #3 (Remarks to the Author):

I have read the revised manuscript, and the referee response. Some of the issues are addressed but not all of them.

I withdraw my complaint about novelty with respect to the work of Prawiroatmodjo et al, which was published only recently. It is properly referenced in the revised manuscript.

Our reply: We are glad to learn that the reviewer has withdrawn his/her complaint about the novelty. In particular, because he/she had made it in his previous report the central part of his negative recommendation towards *Nature Communications* by stating: "*I don't think that this paper rises to the level of novelty or significance required for publication in Nature Communications.*"

It is a bit unfortunate to learn from the 2nd part of the response-line that the reviewer had based this judgment predominantly on the independent work of Prawiroatmodjo et al.

In contrast, we believe that a claim to novelty reaches beyond one particular publication on LAO/STO, as is clear from our reply to the Reviewer 1.

The experimental work is strikingly similar even though the focus is "different". The preset (present?) authors also appear to be working with a "charge puddle" of unspecified origin.

Our reply: In this comment the reviewer returns to the particular publication of Prawiroatmodjo et al., and criticizes both publications implicitly by highlighting that we have not specified the origin of the charge puddle.

Nevertheless, we very much appreciate that the reviewer acknowledges the differences and the similarities of our experiments and those reported by Prawiroatmodjo et al. Indeed, in both cases a 'charge puddle' appears when the two-dimensional superfluid is depleted electrostatically. The reviewer appears to be concerned about this observation and is puzzled by its origin.

Perhaps, it is worth repeating partially what we wrote in our reply to Reviewer 1. The emergence of an electronic state with a non-uniform electron density is a quite common feature for 2-dimensional systems in which the electron density is reduced to zero leading to a metal-insulator transition like in field-effect transistors and also in materials like graphene. Similarly, this is also conjectured to occur in systems where a superconductor-insulator transition occurs. In our view it is quite clear that the charge puddle arises because we are depleting the 2-dimensional electronic system. Because it is dependent on arbitrary material imperfections, the details are unknown and have to be inferred from the experiment. So, we believe the origin is quite clear, its precise dimensions and location are not known but need to be clarified in future experiments, taking into account sample-to-sample variations.

Changes made: We have made more specific in the discussion (page 4) that the occurrence of charge puddles is experimentally expected around the superconductor-insulator phase transition in a 2-dimensional superconductor and we have included references [34] Gantmakher and Dolgoplov, *Physics-Uspekhi* 53, 1 (2010) and [35] Kapitulnik, Kivelson and Spivak arxiv:1712.07215 (2017).

It is nice to see that there is a second device now in the supplement. But there is very little discussion of the data, and there are many qualitative differences between the data that is well described in the main text and the different device in the supplement. The authors describe four regimes: (iii), (ii), (i), (o). I would say that regime (iii) looks similar in the second device, but the resistance plot (Fig.2) in regime (ii) looks quite different from Fig. 12(ii). Even (iii) does not resemble the main text but there is some indication of coulomb diamonds forming. It's not clear why the other regimes were not shown. Also, the diamonds are offset in Fig. 12. The authors should comment on this feature. The second device seems to reach a zero-resistance state, while the device in the main text looks to have a ~ 1 kOhm background resistance. That difference should also be commented upon.

Our reply: We are grateful that the reviewer acknowledges the additional data set. We agree with the reviewer that there are similarities but also differences with the data shown in the main text. However, differently from the reviewer, we believe that certain deviations between different devices are not surprising, but instead are to be expected. This is because, as indicated above, we consider the emergence of non-uniform electron densities an intrinsic part of the physics of these devices. They manifest themselves now in the point-contact configuration, which implies that the pattern at low densities becomes sample-specific. We agree with the reviewer that this is not a trivial challenge. For the time being, we have approached this challenge by clearly labelling the different regimes. For each of the regimes we have addressed the dominant physical processes and conditions. The fact that the devices do not have identical behaviour, but have comparable regimes is for us reassuring that the identification of the regimes is right, which is one of our main claims. In each of the regimes parameters can easily be different leading to differences in the details, such as the ones pointed out by the reviewer. However, interpreting those differences in too much detail would mean a high degree of speculation and would not yet provide us with a well-justified scientific claim. Our choice as authors of this manuscript has been to not over-interpret the available data, and restrict the claim to what can be said safely.

A few more comments about the detailed observations by the reviewer:

a; The reviewer asks why we have not shown the other regimes. The answer is that not for all devices were all regimes accessible within the available gate voltage range.

b; The reviewer states that the 'diamonds' are offset in Fig.12. Unfortunately, the reviewer does not specify what kind of offset he/she is referring to. We assume that the reviewer means the difference in range of gate voltages, compared to the figures in the main text (i.e. "offset" along the gate voltage axis). This is a result of the gate history, which leads to a reduced gate action when the gate voltages are swept often over a wide range.

c; The reviewer asks about the differences in the appearance of the noise-rounded Josephson current by comparing the two devices. Our answer is, as included in the main text, that the actual current-voltage characteristic is the result of the Josephson-effect, with a coupling energy connected to the critical current, as well as the (carrier density related) energy gap, and compared to the thermal energy at the bath temperature. Device 2 has a higher critical current, which suggests more modes in the constriction, consistent with a higher carrier density and a higher gap. This difference is one of the interesting aspects of our experiment because it shows the electrostatic tunability of the Josephson coupling energy through the

gate voltage, while the thermal energy stays the same, causing a change in the appearance of the observable supercurrent branch in the I,V curve.

Changes made: We have made the appropriate comments in the Supplementary, section IX, while referring to the main text, wherever appropriate.

The authors place great importance on having created a split gate geometry. The very first sentence of the abstract reads:

"One of the hallmark experiments of quantum transport is the observation of the quantized resistance in a point contact formed with split gates in GaAs/AlGaAs heterostructures."

However, the authors have not reached this "hallmark". The authors provide no data that would support the existence of QPC behavior as first demonstrated by van Wees back in 1988. Where are the conductance steps?

Our reply: The reviewer criticises that our measurements do not reveal a Van Wees-type staircase with quantized conductance steps. The reviewer appears to consider their absence an especially important aspect when he/she states '*the authors have not reached this "hallmark"*' (referring to the Abstract where we call the original experiments by Van Wees et al. "One of the hallmark experiments of quantum transport").

We are very grateful to the reviewer for raising this question, because it allows us to clarify the commonalities and the differences between an ordinary two-dimensional electron gas of non-interacting electrons, which gives the Van Wees steps, and a superconducting 2-dimensional electron gas. Our paper is clearly focused on the latter.

Let us start by pointing out that the analysis we provide in the manuscript and in the supplementary material is in our view fully sufficient to support the claim of having created a *superconducting quantum point contact* with split gates. We obtain good agreement with the expected behaviour in that regime. It is obvious that the Van Wees *conductance* steps are not expected to occur in a quantum constriction which is *superconducting* because such a system is in a macroscopic quantum state and thus exhibits an infinite conductance, which leaves no room for steps. Therefore, one can no longer use the beautiful simplicity of the normal state behaviour. Instead, one has to re-consider the quantum point contact problem from the perspective of the superconducting state, taking into account the physics of Josephson-junctions. From this perspective other properties need to be analysed, such as the critical supercurrent, as we have done in our analysis in the manuscript, with a consistent outcome.

In principle, we might be able to see a stepwise decrease of the critical current (in Fig.2a). However, the first thing to remember is that given our Fermi wavelengths and the width of the channel we are unavoidably in a few mode regime. In addition, we observe fluctuations in the amplitude, which are a natural by-product of quantum transport with finite elastic scattering.

Having said that, the reviewer may wonder why no conductance steps are visible for currents larger than the critical supercurrent of the junction. This is a relevant question which initially also puzzled us. We realized, however, that this is indeed plausible. This can be understood when recalling two important aspects.

1. **Conductance steps at finite bias:** Quantum transport measurements with normal electrons are typically carried out in the small-signal limit, sometimes also called the linear-response regime. This ensures that the transport energy scale is smaller than the relevant energy scales of the system (such as the energy separation of the modes inside a QPC). Only in this regime are quantized conductance steps observed in a normal QPC. For larger energies the steps are known to drift away from the quantized values and smear out [Patel et al. PRB 44, 13549 (1991)]. This becomes especially relevant when the mobility is not as high as in the technologically well-controlled GaAs/AlGaAs heterostructures. Turning to our data, the voltage scale in figure 3 a) in the main text (and figure 3 in the supplementary) tells us that far above the critical supercurrent transport takes place at energies of 100 to 200 μeV . Comparison with GaAs/AlGaAs devices, where the QPC level spacing is typically of the order of 1-2 meV, and taking into account that the effective mass in LAO/STO is about one order of magnitude larger, we arrive roughly at an energy scale of a few 100 μeV . It is therefore of the same order as the energy scale of transport for currents far above the critical current in our sample. We therefore can no longer safely assume to be in the linear-response regime of small signals. Hence, it is understandable that no clear steps occur here.

We point out that this is also consistent with the measurements reported by Gallagher et al. (Nat. Phys. 10 748 (2014)) who observed a normal state QPC with quantized conductance in doped SrTiO₃. Here we can see (Fig.3a therein) that the steps which are visible for small bias quickly disappear when the bias voltage is increased to approximately 100 μV , for the same reasons.

2. **Number of modes in our constriction:** For the given width between the split gate tips in our experiment and for the given Fermi wavelength in LAO/STO, our SQPC is in a regime where one would expect only 1 or 2 modes inside the constriction (see main text). Therefore, one should not expect an extensive staircase as in the Van Wees experiments. For our QPC we can expect at most only one step, provided that we start with two modes in the channel. This step should occur when the number of modes is reduced from 2 to 1. When the modes are reduced further, from 1 to 0, no step is expected because this transition corresponds to pinch off which shows up in our experiment as regime (ii) where the charge puddle forms. Accordingly, if we start out with only a single mode, we would not expect to see steps at all but only a transition to regime (ii).

Maybe it is worthwhile mentioning here that even in GaAs/AlGaAs the few mode regime, which we have reached also in our experiment, is precisely the regime where QPCs tend to show deviations from the simple conductance step behaviour. The 0.7 anomaly, which appears in the pinch off regime, is one known tantalizing problem which is presumably due to electron-electron interactions. This illustrates that the deviations from a standard free-electron Schrödinger picture are the regime where new, interesting phenomena appear. Our systems is first of all superconducting, a particular kind of electron-electron interaction, the role of the spin is not yet clear and superconducting quantum dots appear, which is new. All of this one could not reveal before, but it becomes apparent due to the split gate configuration.

Changes made: We have added a note to the main text (page 3) on the absence of steps in the conductance, and we have added references to the work by Patel et al. [23] and Gallagher et al. [11].

Instead of observing the characteristic features of a QPC, the behavior seems to be more related to a quantum dot that is not part of their design.

Our reply: We are a bit puzzled about this comment of the reviewer. We have carried out an experiment on the very peculiar, unconventional 2-dimensional superconductor LAO/STO. We claim that we observe in a certain regime quantum dot behavior, which we attribute to the cross-over to an insulating state as expected for a superconductor-insulator transition (regime iii). We also claim that for higher carrier densities further away from the superconductor-insulator transition we are in the superconducting point contact regime (regime i). In our view these two should not be contrasted with each other but are part of the same experiment that we have carried out and reported by clearly labelling the different regimes.

The physics of a few mode superconducting channel is interesting but there have been other reports of narrow superconducting devices, although not with the split gate.

Our reply: We agree with the reviewer that the physics of a few mode superconducting channel is interesting. We like to add that we have shown experimentally that it is a very rich field in which many regimes play a role. The good thing of the split gate technique is that one can get access to them in a single system.

Regarding the reviewer's criticism "there have been other reports of narrow superconducting devices, although not with the split gate" we refer him/her to our reply to Reviewer 1, where we have explained this issue in great detail.

It is not clear how the split gate is acting on the system or how it is modifying transmission through the device.

Our reply: We appreciate the critical comments from the reviewer. In this particular case, however, we are puzzled.

Generally speaking, in split gate experiments the gate couples capacitively to the electronic system underneath. Thus, a negative potential on the gate reduces the charge density in the electronic system under the gate and eventually also in the region between the tips of the split gates, thus modifying the constriction. From this perspective it becomes obvious how the transmission through the device is modified.

We believe that this picture also applies to our experiments.

The main issue that I have still is the impression that the manuscript gives that the split gates are somehow directly responsible for the behavior. There is no geometric feature in either device gate that can be said to produce a quantum dot, and yet much of the data seems to be defined by this quantum dot behavior.

Our reply: We take note of the fact that the reviewer brings up the 'main issue'.

Perhaps we should repeat what we have said above. By reducing the electron density, the LAO/STO system approaches the insulating state from a superconducting state. This is carried out in our experiment locally at the narrow channel by means of the split gates. Therefore we see the various regimes. This is providing information about the process of transition to the insulating state, from a superconducting quantum point contact regime towards a non-uniform regime involving the clear signatures of a local charge puddle manifesting itself with all the characteristic features of a quantum dot and which even shows signatures that it is a charge puddle which in itself is superconducting.

Such a behavior does not arise from the geometric configuration but is an unavoidable intermediate state for a low carrier density 2-dimensional electronic system. The split gates enable us to reveal this intermediate state through this transport experiment, which is otherwise very difficult to access.

The simulation that has a rather large mysterious quantum dot in between the gates.

Our reply: We hope that our explanation above on the emergence of charge puddles as an unavoidable intermediate state in the transition from a superconductor to an insulator or even for a metal-insulator transition, has taken away the reviewer's concern about the perceived "mysterious" quantum dot.

And not evidence of ordinary van Wees QPC behavior is provided.

Our reply: see above.

In summary, my concern is that the paper as written is highly misleading. I cannot recommend publication in this journal.

Our reply: We very much appreciate the concern of the reviewer. We believe however, that (1) we have pointed out very precisely the different regimes, with their underlying physics, and (2) we have made very clear, also in the manuscript, that the emergence of the charge puddle is not a direct result of the gate geometry but it reflects a property of our two dimensional superconductor.

In our view this leaves no reason to be afraid that the manuscript is misleading.

We do agree that we could modify the title towards 'Transport regimes of a superconducting quantum point contact with split gates in the two-dimensional LaAlO₃/SrTiO₃ superfluid', because the present title appears to imply only one regime. The abstract however makes already clear that we take a wider scope. We would like to leave it to the Editor to recommend such a change.